# Feasibility and effect of high-intensity training on the progression of motor symptoms in adult individuals with Parkinson's disease: A systematic review and meta-analysis

**Ingrid Garcia De Sena[1], Adalberto Veronese da Costa[2], Isis Kelly dos Santos[2,3], Dayane Pessoa de Araújo[4], Francisca Tayná da Silva Gomes[1], José Rodolfo Lopes de Paiva Cavalcanti[4], Maria Irany Knackfuss[2], Micássio Fernandes de Andrade[5], Paloma Katlheen Moura Melo[1], Ivana Alice Teixeira Fonseca [2,6]***

1 Multicenter Postgraduate Program in Physiological Sciences, Health Sciences College, State University of Rio Grande do Norte (UERN), Mossoro, Brazil, 2 Department of Physical Education, Postgraduate Program in Health Society, UERN, Mossoro, Brazil, 3 Departament of Physical Education, Federal University of Rio Grande do Norte (UFRN), Natal, Brazil, 4 Laboratory of Experimental Neurology, Multicenter Postgraduate Program in Physiological Sciences, Nursing College, UERN, Mossoro, Brazil, 5 Postgraduate Program in Health Society, Health Sciences College, Molecular Biology Laboratory, UERN, Mossoro, Brazil, 6 Department of Physical Education, Multicenter Postgraduate Program in Physiological Sciences, UERN, Mossoro, Brazil

* ivanateixeira@uern.br

## Abstract

### Background

To determine the feasibility and effect of high-intensity interval training (HIIT) in individuals with Parkinson's and their effect on symptom modification and progression.

### Methods

We conducted this systematic review following the Preferred Reporting Items for systematic review and meta-analysis (PRISMA). All studies were searched in seven databases: MEDLINE (PubMed), Cochrane Central Register of Controlled Trials, Web of Science, EMBASE, SPORTDiscus, Virtual Health Library (VHL) and SCOPUS in September 2020 and updated in June 2023. The risk of bias was assessed by the Cochrane Collaboration tool and Grading of Recommendations Assessment, Development and Evaluation (GRADE) tool. We used standardized mean difference (SMD) with a 95% confidence interval (CI) and random effects models, as well as the non-parametric Cochran's Q test and $I^2$ inconsistency test to assess heterogeneity.

### Results

A total of 15 randomized clinical trials with 654 participants (mean age, 65.4 years). The majority of studies included high intensity training interventions versus moderate intensity, usual care, or control group. The meta-analysis comparing high-intensity exercise versus control group showed an improvement in the disease severity (MD = -4.80 [95%CI, -6.38;

**Data Availability Statement:** All relevant data are within the paper and its Supporting Information files.

**Funding:** The authors received no specific funding for this work.

**Competing interests:** The authors have declared that no competing interests exist.

-3.21 high evidence certainty); maximum oxygen consumption (MD = 1.81 [95%CI, 0.36; 3.27] very low evidence certainty) and quality of life (MD = -0.54 [95%CI, -0.94; -0.13] moderate evidence certainty). The results showed that high-intensity exercise compared with moderate intensity exercise group showed a improve motor function and functional mobility measured by the TUG test (MD = -0.38 [95%CI, -0.91; 0.16] moderate evidence certainty) with moderate heterogeneity between studies.

## Conclusion

High-intensity exercise performed in both continuous and interval modes when compared with control groups may provide motor function benefits for individuals with Parkinson's disease. HIIT may be feasible, but the intensity of the exercise may influence individuals with Parkinson's disease. However, there was a lack of evidence comparing high intensity and moderate intensity for this population, as the results showed heterogeneity.

## Introduction

Parkinson's disease (PD) is the second most common chronic neurodegenerative disorder in older adults, affecting about 1% of the world's population over 60 years of age [1]. It is characterized by degenerating dopaminergic neurons in the substantia nigra pars compacta (SNpc), with a consequent dopamine deficit in the synapses of the central nervous system [2–4]. As a result of neurodegeneration, deficits such as tremor, muscle stiffness, bradykinesia, autonomic dysfunction, pain, and sleep disorders compromise the functional performance and quality of life of Parkinson's patients, increasing their risk of morbidity and mortality [4–7]. There are several treatments for Parkinson's disease including medicines, surgical treatment, and other therapies to increase the level of dopamine in the brain and help control non-movement symptoms. In addition, other therapies may manage the symptoms and modify the course of the disease and protect neurons against damage resulting from neurodegeneration [8–10]. In addition, therapies based on strength, cardiovascular and/or stretching exercise improve balance, flexibility, cardiorespiratory fitness, muscle strength, cognition, and quality of life in individuals with PD.

Studies highlight the potential benefits of low to moderate intensity exercise (37% to 63% of maximal oxygen consumption—VO2 max) for people with PD[10–14]. From this perspective, some issues have arisen about how exercise intensity affects its effectiveness to delay motor symptoms as disease progresses [9,15,16].

According to studies, exercise intensity is often measured using methods such as heart rate (HR), heart rate reserve (HRR) or oxygen uptake (VO2). There are several types of exercises and intensities; additionally, high-intensity is defined and categorized as vigorous effort (70%–90% of peak HR; 60%–84% of HRR; 60%–79% of peak VO2) or "very hard" effort ($\geq$ 90% of peak HR; $\geq$ 85% of HRR; $\geq$ 80% of peak VO2) [17–19]. Moreover, the popular high-intensity interval training (HIIT) is interpreted as an exercise protocol which includes high-intensity exercise bouts interspersed with rest periods [20]. Evidence regarding PD suggests that these more intense exercises, above 70% of VO2peak, can induce activity-dependent neuroplasticity and produce more effective effects in delaying motor dysfunction when compared to low (37%-45% VO2) or moderate intensity exercises (45%-63% VO2) [16,21–23].

High-intensity exercise training may have greater effects on health parameters in obese and hypertensive patients when compared with moderate intensity [24,25], but the inclusion of

higher training intensities as an adjuvant treatment for PD is not fully understood; especially considering that the motor limitations caused by the progressive symptoms already observed from stage II of the disease can impair the feasibility of performing the exercise, as is the case of those performed on a treadmill.

However, there is no consensus for prescribing the exercise intensity in PD, and variations in training protocols may have different effects, changing the physiological response of individuals to exercise [26,27]. Based on possibility of delays in symptom progression of the disease and the consequent improvement in functionality and quality of life that high-intensity physical exercise can present, and since there are no systematic reviews that we know of which have investigated this training modality in individuals with Parkinson's disease, this study aims to review the evidence about the feasibility and effects of high-intensity exercise for people with Parkinson's disease.

## Methods

### Study registration

This systematic review and meta-analysis were developed following the recommendations of Cochrane guidelines [28]. All report and preparation processes followed the Preferred Reporting Items for Systematic Reviews and Meta-Analysis (PRISMA) guidelines [29]. The protocol for the construction stages of this review was registered in the International Prospective Register of Systematic Reviews (PROSPERO), under registration number CRD42020188473.

### Eligibility criteria

We included randomized controlled trials following these criterias: **Population:** adults and older adults diagnosed with Parkinson's disease; **Intervention:** exercise interventions classified as high-intensity; **Comparator:** moderate or low exercise or usual care (control group); **Outcomes:** The primary outcomes were progression of motor symptoms, as measured by the UPDRS, and feasibility of high-intensity physical exercise, as measured by adherence to training, participation rate until the end of the study, and achievement of the previously established target heart rate. Secondary outcomes: lower limb functionality measured by the TUG test, maximum oxygen consumption, and quality of life measured by the PDQ-39. We excluded studies which involved animals; systematic reviews and meta-analysis; dissertations; and theses.

### Search methods for identification of studies

We performed a search in seven databases: Medline (PubMed), Cochrane Library, Web of Science, EMBASE, Library virtual of Health (BVS), SportDiscus and SCOPUS. No language or year restriction for articles were considered. The entire search process occurred in November 2020, but it was updated in June 2023. The search for descriptors used was performed in consultation with the Medical Subject Headings (MeSH), through the site of U.S. National Library of Medicine (NLM); Descriptors in Health Sciences (DeCS), and Embase Subject Headings (Emtree). We combined the subject, terms, synonyms, and keywords for "Parkinson's disease" and "High-Intensity training" in the search strategy using the Boolean operator AND/OR. The search strategy is presented in Appendix A in S1 Appendix.

### Study selection

Two authors screened the titles, abstracts and full texts independently (IGS and PKMM) using the Rayyan QRCI program [30]. All eligible studies were identified using the inclusion criteria.

All duplicates were excluded after studies had been selected based on their titles and abstracts. Only full texts were evaluated to be considered potentially eligible. Any conflicts were resolved by a third reviewer (IATF).

## Data extraction process

Two authors extracted all information independently using a template. We used standardized data extraction according to the following items: data on participants' characteristics (number of participants, age, and PD stage), types of interventions (experimental and control group), outcomes measured and other details. A third reviewer (IATF) checked all data and ensured the process quality, and any differences were resolved by discussion and consensus. The entire process was conducted using the Rayyan QCRI program.

## Risk of bias

We used the Cochrane Collaboration tool to assess the risk of bias (Cochrane Risk of Bias Tool) for the methodological quality of the studies included in this review [31]. According to this tool, each study was assessed independently based on the following seven domains: random sequence generation, allocation concealment, blinding of participants and professionals, blinding of outcome evaluators, incomplete outcomes, selective outcome reporting and other sources of bias. A response was generated for each of these domains based on three different categories: the high risk of bias, the risk of uncertain bias and the low risk of bias. Two authors independently (IGS and DPA) assessed risk of bias. When necessary, a third reviewer resolved disagreements (IKS).

## Data synthesis

We planned to perform the meta-analysis comparing high intensity vs. moderate/low intensity exercise programs. We implemented standardized mean difference (SMD) with a 95% confidence interval (CI) and fixed-effect models for continuous results. The non-parametric Cochran's Q test, which verified whether the training had identical effects, and the $I^2$ inconsistency test were used to assess heterogeneity [32]. The p-value of 0.10 was used to indicate whether the heterogeneity was significant. The heterogeneity magnitude was assessed by calculating the $I^2$, which ranges from 0% to 100%, indicating substantial heterogeneity when greater than 50% and considerable heterogeneity when greater than 75% [33]. A random effects model was used whenever there was a high degree of heterogeneity between studies [34]. The fixed effects model was used to estimate the change from baseline $I^2 > 50\%$ to explain the intervention effect size within the group. The studies were grouped according to the variables analyzed, considering high-intensity training versus moderate intensity training or high-intensity training versus the control group. The Review Manager software program (RevMan® 6.0) was used for all analyses.

## Assessment of the quality of evidence

We used the Grading of Recommendations Assessment, Development and Evaluation (GRADE) tool to provide the quality of evidence and the strength of health recommendations for the intended outcomes[35]. The level of evidence in the GRADE method represents the confidence in the information that was used in the review, being performed for each analyzed outcome using the available set of evidence. The quality of evidence is classified into four levels: high, moderate, low and very low [36]. The data regarding the GRADE tool are described in Appendix B in S1 Appendix.

## Results

### Description of studies

A total 3,159 articles were found, but 1,366 were removed as duplicates. Next, 1,793 studies were evaluated in the screening stage by title and abstract for potential eligibility, with 1,723 excluded for not being thematically adequate. Thus, 70 studies with selection potential for full text reading were identified, and 55 studies were then excluded according to the eligibility criteria; as a result, 15 studies were selected for the review (Fig 1).

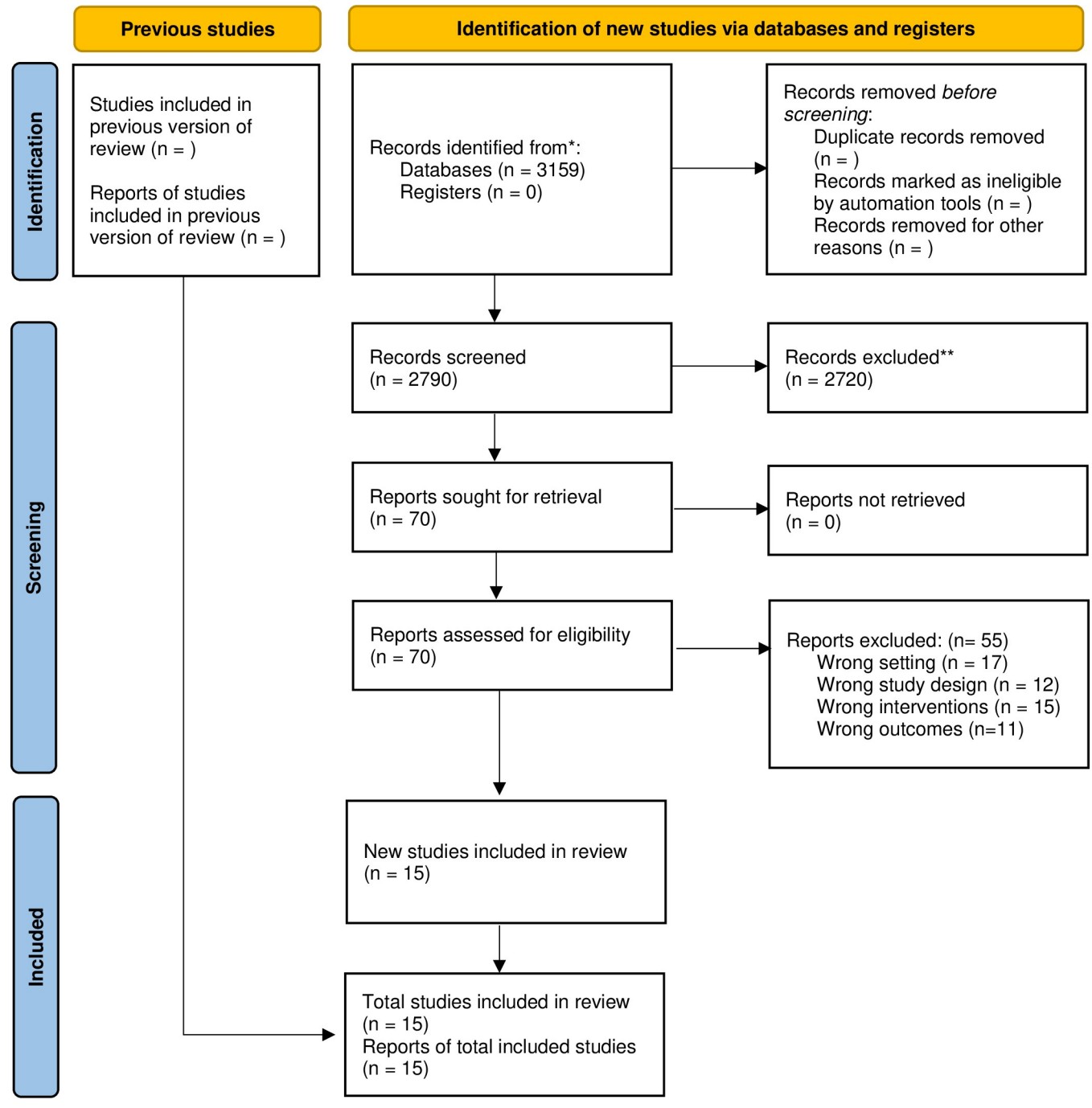

**Fig 1. Flowchart of studies included.**

**Table 1. Characteristic of studies included (n = 15).**

| Author, year | Country | Duration (week) | Design of study | Sample size | Participants M/F | Age (years) | Age Mean (SD) (y) | Classificação de Hoehn e Yahr |
|---|---|---|---|---|---|---|---|---|
| Cancela et al., 2020 [50] | Spain | 8 weeks | RCT–pilot | 14 | 12/2 | 64–72 | 68.33 (4.40) | 1–3 |
| Duchesne et al., 2015 [51] | Canada | 12 weeks | RCT | 39 | 21/18 | 40–80 | 59 (7.11) | 1–2 |
| Duchesne et al., 2016 [52] | Canada | 12 weeks | RCT | 39 | 21/18 | 40–80 | 59 (7.11) | 1–2 |
| Ergun et al., 2014 [53] | USA | 24 months | RCT | 60 | 41/19 | 50–80 | 65.4 (6.2) | 1–3 |
| Fernandes et al., 2020 [54] | Brazil | 12 weeks | RCT | 29 | 13/8 | 60–77 | 68.6 (8.3) | - |
| Fiorelli et al., 2018 [55] | Brazil | 3 weeks | RCT | 12 | 6/6 | 53–81 | 66.5 (8.0) | 1–3 |
| Fisher et al., 2008 [56] | USA | 8 weeks | RCT | 30 | 13–17 | | | 1–2 |
| Harvey et al., 2019[57] | UK | 12 weeks | RCT | 20 | 12/8 | 61–65 | 68.5 (6.82) | 1–3 |
| Moberg et al., 2014 [58] | Denmark | 32 weeks | Clinical trial | 24 | 9/15 | 61–62 | 61.3 | 1–2 |
| O'Callaghan et al., 2019 [59] | UK | 12 weeks | Clinical trials | 52 | 26/26 | 55–86 | 74.4 (7.2) | 1–3 |
| Schenkman et al., 2018 [60] | USA | 26 weeks | RCT | 128 | 73/55 | 40–80 | 64.0 (9.0) | 1–2 |
| Segura et al., 2020 [61] | Columbia | 8 weeks | RCT | 13 | 7/6 | 56–65 | 57.8 | 1–3 |
| Shulman et al., 2013 [62] | USA | 12 weeks | RCT | 67 | 50/17 | 42–86 | 65.8 (10.7) | 1–3 |
| Tollár et al., 2018 [63] | Hungary | 5 weeks | RCT | 74 | 36–38 | 62–75 | 70.0 (4.69) | 2–3 |
| Tollár et al., 2019 [64] | Hungary | 3 weeks | RCT | 55 | 29/26 | 63–70 | 67.6 (3.75) | 2–3 |

Note: F (female); M (male); Classificação de Hoehn e Yahr*; RCT–randomized Controlled trials.

### Characteristics of the included studies

The majority of studies were randomized controlled trials (RCTs) conducted in: Canada (2 studies) [37,38]; United States (4 studies) [23,26,39,40]; Hungary (2 studies) [41,42]; Brazil (2 studies) [43,44]; United Kingdom (2 studies) [45,46]; Spain [47], Denmark [48], and Columbia [49] (one each). The participants of the studies were recruited from clinical health centers at universities, hospitals, and clinics in several countries. The sample sizes range between 10 and 128 participants, mean average age was 65.4 years and ranged across studies from 30 and 86 years, men and women, to whom the mild (1–2), moderate (3) and severe (4) stages of Parkinson's disease were attributed by the Hoehn and Yahr classification, with only one study which did not report the participants' Hoen & Yahr classification stage [44]. Regarding adjustments for confounding factors, 14 studies (48.2%) showed a high variation in the age of the participants, which is a potentially modifying factor of symptoms and responses to training (see Table 1).

### Characteristics of interventions

The interventions varied widely among the 15 studies. Each study included high-intensity exercise protocols in different modalities, including bike training, stationary recumbent bike, walk, jogging and treadmill. Some studies combined HIIT and strength, power, flexibility, balance, cardiorespiratory endurance, and functional training. The majority of studies compared high-intensity training with moderate-intensity training, control group or groups that performed only usual care[56–58,60–64], classified as low-intensity training, and there was additionally a combination of high intensity training with strength training and balance; only one study did not compare with another group.

A total of 15 studies used maximum heart rate ($HR_{max}$) to assess training intensity, characterizing high intensity training when the values were above 70% $HR_{max}$. Only two of the studies measured the training intensity with the Borg Rating of Perceived Exertion scale [65], in which values above 15 characterized high-intensity training.

The training period length varied widely across the 15 studies; six studies had 12 weeks of training with three sessions per week. Two studies used 8 weeks of training, 24 months, 32 weeks, 26 weeks, 3 weeks, 5 weeks, 24 sessions and 3 sessions (each study). The average weekly frequency ranged from two to three days, and session duration ranged from 25 to 60 minutes of training. The intervention characteristics are described in Table 2.

## Characteristics of outcomes

The feasibility of high intensity training was analyzed by adherence to training percentages and by reaching the previously determined target maximum heart rate ($HR_{max}$). The $HR_{max}$ target reached between the studies ranged from 61% to 100%, and these values refer to the study with the lowest and the highest rates of volunteers who reached the expected intensity, respectively. The results regarding adherence to training ranged from 66% to 100% participation until the end of the study, with losses reported by dropouts (ranged from 1 to 8) not related to the training protocol, such as changing the city and death from causes not related to the exercises.

Motor function assessed by the motor subscale of the Unified Parkinson's Disease Rating Scale (MD-UPDRS) was used by 11 studies, and the assessment of maximum oxygen consumption (VO2) was measured by six studies using cardiopulmonary exercise testing system (COSMED Srl, Rome, Italy).

The results of the studies related to improvement in motor scores on the MD-UPDRS show that high-intensity training can delay the progression of motor symptoms in Parkinson's disease. Other parameters which also influence disease progression and that are related to performance showed significant improvement, such as $VO_2peak$, mobility and functionality of the lower limbs measured by Time Up and Go (TUG) test [66] was included in four studies, and cardiorespiratory fitness assessed by the 6-Minute Walk Test (6MWT) was used in two studies. Furthermore, other important findings such as improved quality of life evaluated by the Parkinson's Disease Questionnaire (PDQ-39) [67] was included in five studies and increased levels of circulating BDNF in people with PD after a high-intensity training period were also observed.

## Risk of bias in included studies

The risk of bias of all the studies (by domain items) was found as follows: all studies were classified as low risk of bias for random sequence generation (randomization), since they allocated their participants at random and described how this allocation was performed; one study was at high risk of bias for the allocation concealment item (which describes whether the study was adequately randomized), because they only used a random list of numbers generated by an open randomization process; one study had a high risk of bias in blinding the participants and professionals because they did not describe how the double-blind study proceeded, nor did the study have blinding in the assessments performed during the research, which made the outcome susceptible to influence from the lack of blinding; only one study had a high risk of bias for the item blinding of outcome evaluators, because there was no blind assessment of the outcomes, and the assessed outcomes are influenced by the lack of blinding; one study was assessed as unclear risk of bias for the item incomplete outcomes and selective outcome report, because not all results of important outcomes were described; lastly, three studies had a high risk of bias and 12 studies had a moderate risk of bias for other sources, both related to the study design (Fig 2).

## Meta-analysis

**High-intensity exercise vs control group.** *Progression of motor symptoms.* The Unified Parkinson's Disease Rating Scale (UPDRS) was used in 11 studies to assess symptom

**Table 2. Characteristic of interventions and results of studies included (n = 15).**

| Author, year | Intervention | Duration Frequency (time/week) | Intensity | Comparison | Intensity | Adherence | Dropouts | Main results |
|---|---|---|---|---|---|---|---|---|
| Cancela et al., 2020 [50] | HIIT/balance training/ cycle ergometer | 1x/week 35 minutes | >70% HRmax | Balance training | | 100% | 2 | • Improvements in: PDQ-39 (p = 0.036); • UPDRS-MD (p = 0.047) for intervention group; • This exercise is effective to reduce the motor symptoms |
| Duchesne et al., 2015 [51] | HIIT, stationary recumbent bike-training program | 3x/week 40 minutes | 80% VO2peak | Continuous bike training | 60–80% VO2peak | ≥75% | NR | • Improvement in: aerobic capacity (VO$_{2máx}$: p <0.001); • Suggests an improvement in physical fitness for individuals with PD |
| Duchesne et al., 2016 [52] | High-intensity, stationary recumbent bike-training program | 3x/week 40 minutes | 80% VO2peak | Continuous bike training | 60–80% VO2peak | ≥75% | NR | • Functional changes in the hippocampus, striatum and cerebellum related to motor learning and improvement in behavioral performance observed in patients with PD |
| Uc et al., 2014 [53] | HIIT/ walk and run | 3x/week 45 minutes | 3' - 70–80% 3'- 80–90% of the HRmax | MICE in a community environment | <70% of the HRmax | 83% | 8 | • Walking in a community environment is safe, well tolerated and improves aerobic fitness, motor function, fatigue, mood, executive control and quality of life in mild to moderate PD |
| Fernandes et al., 2020 [54] | HIIT/walk and jogging/ running | 3x/week 60 minutes | 21' - 15–17 to RPE | MICE/treadmill | 26'- 11–14 RPE | 93.3% | 6 | • Suggests that the intensity of exercise may influence the adaptation induced by intense training on endothelial reactivity |
| Fiorelli et al., 2018 [55] | HIIT/stationaty bike | 3 sessions (Acute training) | 4'- 9–11 on RPE 2'- 11–13 on RPE 1'- 13–15 on RPE | MICE/treadmill | 11–13 RPE | 100% | 2 | • Improvements in: auditory memory, attention; • Suggests that the HIIT was able to provide improvements in cognitive performance in people with PD |
| Fisher et al., 2008 [56] | HIIT/ body weight–supported treadmill training | 24 sessions | >75% of the HRmax | Control group no exercise, only advices | | 100% | none | • Demonstrated post-exercise increases in gait speed, step and stride length, suggesting that high-intensity exercise can normalize corticomotor excitability |
| Harvey et al., 2019 [57] | HIIT/ Speedflex machine | 3x/week 45 minutes | ≥ 85% of the HRmax | Control/ without exercise | | 85% | 3 | • VO$_{2\text{-peak}}$ increased in the both groups; • Technically feasible because the participants reached ≥85% of the maximum HR during training |
| Moberg et al., 2014 [58] | Strength, flexibility, balance and cardiorespiratory resistance training | 2x/week 35 minutes | ≥85% of the HRmax | Control/ Without exercise | | 66%. | 4 | Improvements in improved the UPDRS motor subscores (- 10 points, p = 0.045): activities of daily living (p = 0.006), as well as the items emotional well-being (-11.0 points) and body discomfort (- 7.14) of PDQ39 |
| O'Callaghan et al., 2019 [59] | HIIT/elliptical | 3x/week 45~60 minutes | ≥ 85% | MICT Cardiorespiratory strength and resistance exercises | 60–80% | 100% | 4 | • The HIIT protocol was superior to the MICT for circulating BDNF levels after training, with an increase of 84.2% compared to the baseline |

*(Continued)*

**Table 2.** (Continued)

| Author, year | Intervention | Duration Frequency (time/week) | Intensity | Comparison | Intensity | Adherence | Dropouts | Main results |
|---|---|---|---|---|---|---|---|---|
| Schenkman et al., 2018 [60] | HIIT/ Treadmills exercise | 4x/week 50 minutes | 80–85% HRmax | * Continuous treadmill training *Control without exercise | 60–65% HRmax | 80.2% | 5 | • Improvements UPDRS motor score |
| Segura et al., 2020 [61] | HIIT/ cycling program on a stationary | 3x/week 30~40 minutes | 80% of the HRmax | Control/ Without exercise | | 100% | NR | • Improved peak VO2 (p = 0.041); • UPDRS (-9 points); • Circulating BDNF levels |
| Shulman et al., 2013 [62] | HIIT/treadmill | 3x/week 45 minutes | 70–80% Of the heart rate reserve | *Continuous treadmill training * Strength + stretching exercises | 40–50% | 70–80% | 3 | • Improved the distance at 6MWT (moderate intensity group: 12% p = 0.001, stretching + strength group: 9%; p <0.02 and high intensity group: 6%; p = 0.07) • Both treadmill exercises improved $\dot{V}O_{2-peak}$ 7% to 8%; p <0.05 |
| Tollár et al., 2018 [63] | High intensity agility intervention | 5x/week 50 minutes | > 80% of the HRmax | *Continuous bike training (CYC) *Control without exercise | 80% of the HRmax | 100% | none | • Improvements in UPDRS-II scores. • PDQ-39 and MWT |
| Tollár et al., 2019 [64] | High-intensity agility training | 5x/week 50 minutes | ~80% | *Exercise *Control/ without exercise | | 80% | 1 | • Improve motor sensory agility program was able to delay the progression of motor symptoms in PD |

Note: 6 MWT (6-minute walk test); BAI (Beck Anxiety Inventory); BDNF (Neurotrophic Factor derived from the Brain); CAE (Continue Aerobic Exercise); CYC (Stationary Cycloergometer); EPR (Perceived Effort Scale); EQ-5D (Euro-Quality Of Life-5 Dimensions); EXE (Exergaming); fMRI (Functional Magnetic Resonance); HIIT (High Intensity Interval Training); HR (Heart Rate); HRmax (maximum heart rate); MBT (mini-Balance Evaluation Systems Test); MDS-UPDRS (Society's Disorders Movement Review of the Unified Parkinson's Disease Classification Scale); MICE (Continuous Training of Moderate Intensity); MMSE (Mini Mental State Evaluation); MoCA (Montreal Cognitive Assessment); MVIC (Maximal Voluntary Isometric Force); PD-CFRS (PD Cognitive Functioning Rating Scale); PDQ-39 (Parkinson's Disease Questionnaire– 39); PDSS (Parkinson Disease Sleep Scale); SE-ADL (Schwab and England Activities of Daily Living); SRT (Series reaction time); TMS (Transcranial Magnetic Stimulation); TUG (Time Up And Go); SFT (Senior Fitness Test); UPDRS (Movement Disorders Society Unified Parkinson's Disease Rating Scale); VO2peak (peak oxygen consumption); NR no reported.

progression and delays in the disease, but only 7 studies reassessed the volunteers from baseline changes. This scale is a validated tool to assess the severity of Parkinson's disease, monitoring the progression of symptoms and the effectiveness of the treatments employed. This scale is divided into four parts: (1) mental activity, behavior and mood; (2) activities of daily living (ADLs); (3) motor exploration; and (4) complications of drug therapy [68]. The representation for decreasing the progression of the disease is indicated by the decrease in the scores of each item.

Six studies with 210 volunteers were included to analyze the effect of high intensity exercise versus control group on the progression of motor symptoms observed using UPDRS (Fig 3). The meta-analysis showed an improvement with statistical significance in the disease severity (P< 0.00001), with a balance of -4.80 points in the total scores (95%CI, -6.38; -3.21), which favored the volunteers of high-intensity exercise with high evidence certainty (Fig 3).

**Maximum oxygen consumption.** Cardiorespiratory fitness determines an individual's ability to sustain dynamic exercises of moderate and high intensities for a long period of time,

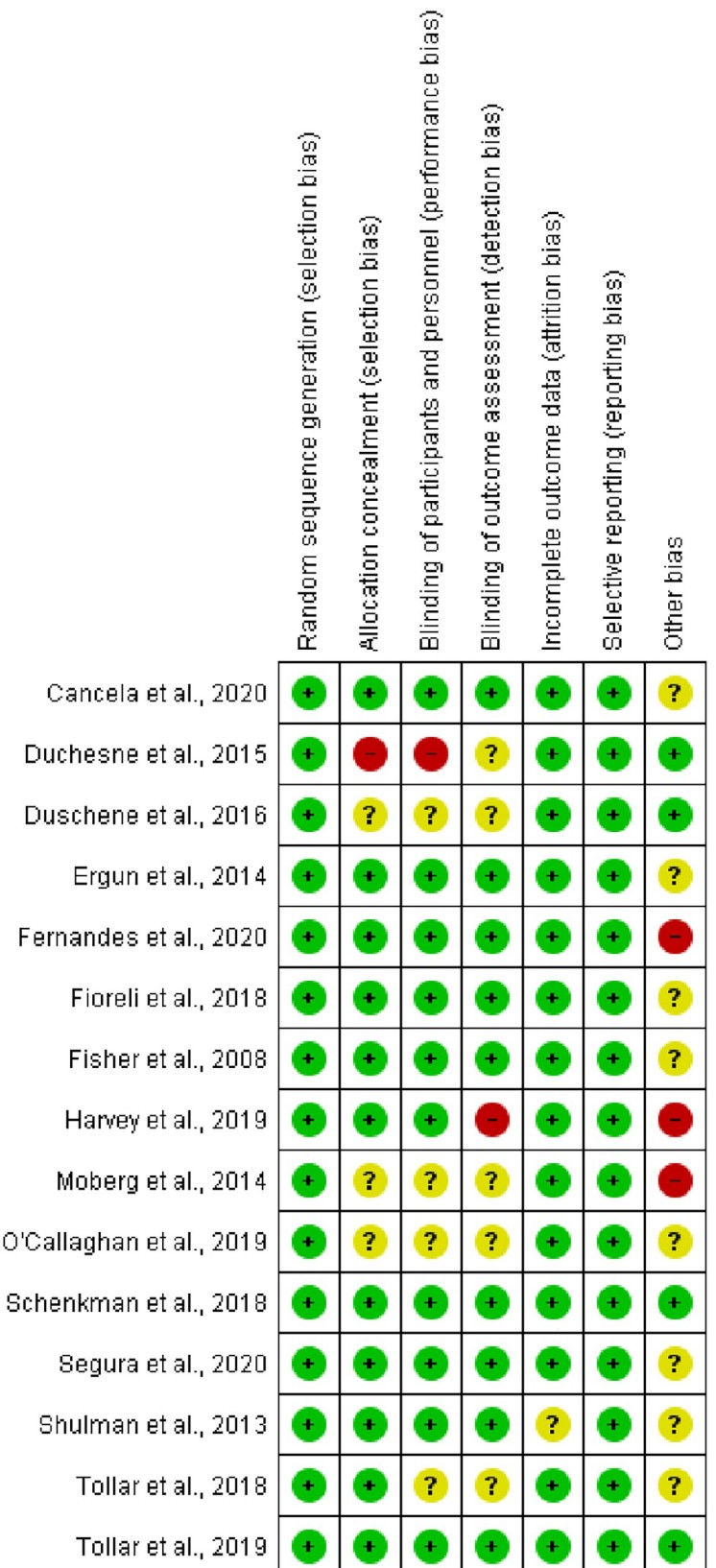

**Fig 2. Risk of bias summary for all studies.**

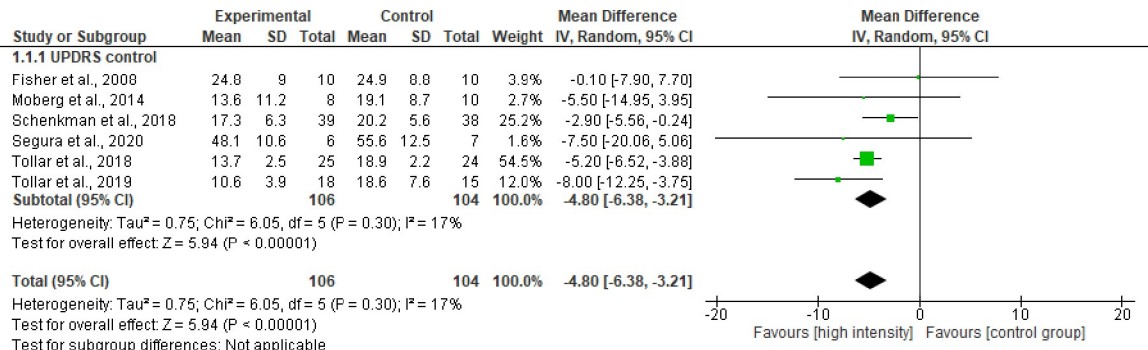

**Fig 3. Effect of high intensity exercise versus control group on progression of motor symptoms.**

in addition to providing information on morbidity prognosis and capacity to respond to treatment in certain pathologies [69]. The current gold standard for analyzing cardiorespiratory fitness is the direct measurement of VO2peak.

Four studies with low risk of bias totaling 164 volunteers were included to analyze the effect of high-intensity exercise versus control group in the assessment of maximal oxygen consumption (VO$_2$peak) (Fig 4). The meta-analysis showed an improvement in statistical significance in VO$_2$peak (P<0.0007), with an increase in oxygen consumption favoring the high-intensity exercise volunteers [1.78 l/m (95%CI, 0.82; 2.75), along with high heterogeneity (82%) and very low evidence certainty.

**Quality of life.** The chronic and progressive nature of PD negatively affects the quality of the daily life activities of PD patients and consequently their quality of life [4]. As an indicator of PD progression and management, quality of life assessment is considered an important result for treating the disease [70]. The 39-item Questionnaire for Parkinson's disease (PDQ-39) is a validated and widely used instrument for this assessment composed of eight scales that assess mobility, activities of daily living, emotional well-being, stigma, social support, cognition, communication, and body discomfort. Improvement in the quality of life is represented by a decrease in the score of each item [67].

Three studies with 100 volunteers were included to analyze the effect of high intensity exercise versus control group on quality of life by the PDQ-39 (Fig 5). The meta-analysis showed an improvement with statistical significance in quality of life (P = 0.001), with a balance of -0.59 points in the total scores (95%CI, -0.94; -0.13), which favored the high-intensity exercise volunteers and showed moderate evidence certainty.

**High-intensity exercise vs moderate intensity exercise.** *Progression of motor symptoms.* Five studies with 219 volunteers were included to analyze the effect of high intensity exercise versus moderate intensity exercise on UPDRS (Fig 6). The meta-analysis showed an

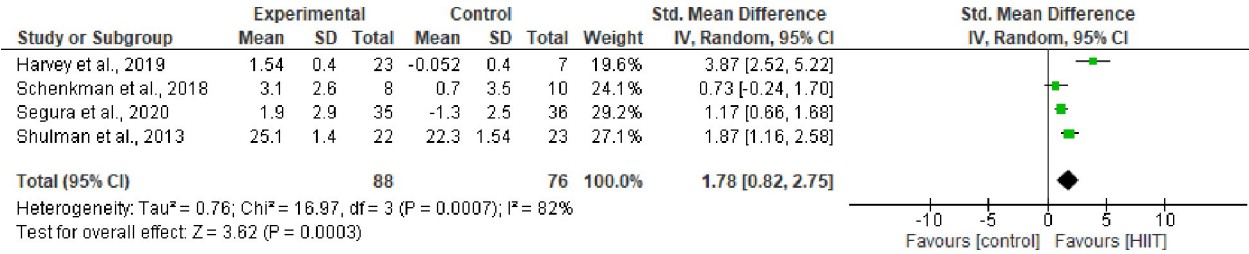

**Fig 4. Effect of high-intensity exercise versus control group on maximum oxygen consumption.**

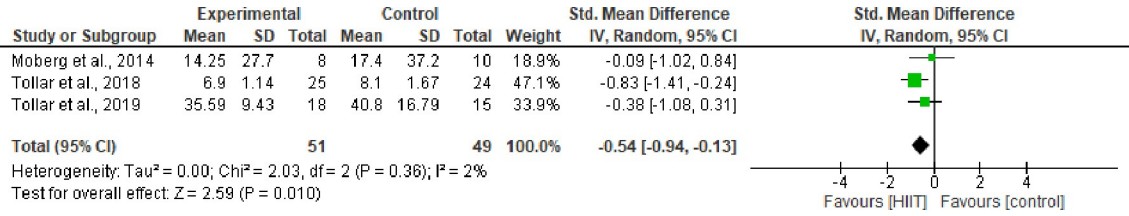

**Fig 5. Effect of high-intensity exercise versus control group on quality of life.**

improvement with statistical significance in the disease severity (P<0.03), with a balance of -2.70 points in the total scores (95%CI, -5.06; -0.33), which favored the high-intensity exercise volunteers (Fig 3), with moderate heterogeneity.

## Functionality mobility

Functional performance measures for people with Parkinson's disease (PD) are necessary and correlate with disease progression, as motor changes caused by progressive disease symptoms resulting from neuronal degeneration are associated with the decline in functional capacity of these individuals [2].

The Time Up and Go (TUG) test was applied to assess basic functional mobility, which analyzed the number of required steps from each individual to perform the activity "getting up from a chair which has arms, walking 3-meters distance and returning to the chair", considering that improvement for the test is indicated by decreasing the time spent to perform the activity, with higher values of time and number of steps representing a greater risk for falls [66].

Three studies with 64 volunteers were included to analyze the effect of high-intensity exercise versus moderate intensity exercise on the TUG test (Fig 7). The meta-analysis showed no statistical significance in functional mobility (P = 0.33), with a balance of -0.38 seconds to perform the TUG test (95%CI, -0.91; 0.16, moderate evidence certainty), when compared to moderate intensity exercises.

## Discussion

Results from this systematic review and meta-analysis indicate that prescription of high-intensity exercise for individuals with PD may be feasible, in which the volunteers reached the previously established HRmax target for exercise, which can be considered a good adjuvant treatment of the disease to improve symptoms. However, these findings need more investigate

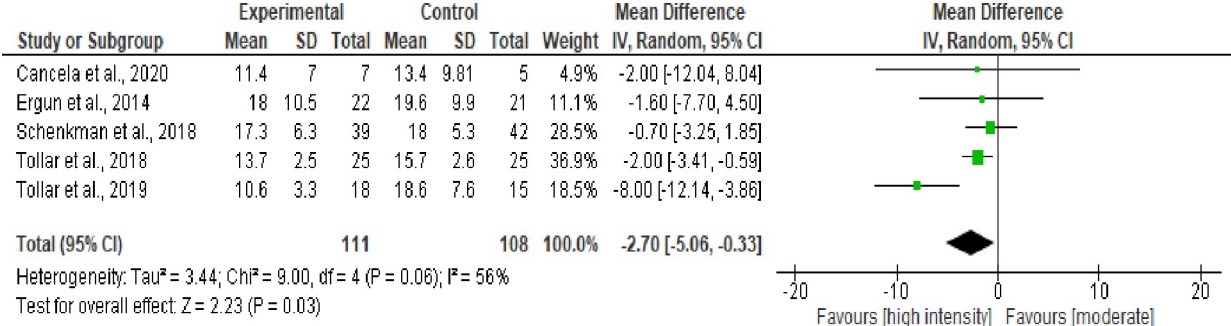

**Fig 6. Effect of high-intensity exercise vs moderate intensity on the progression of motor symptoms.**

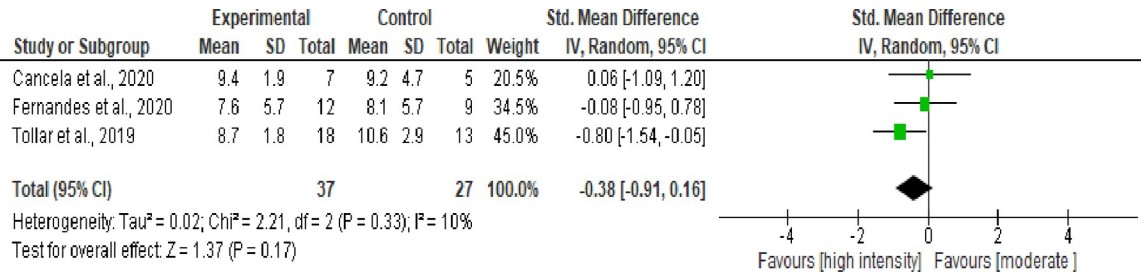

**Fig 7. Chart for the effect of high-intensity exercise versus moderate intensity on functional mobility (TUG).**

about adherence and adverse events, a few studies reported this information with details. The analysis of the effect of the interventions identified potentially changes in motor symptom and functional mobility after HIIT performed in continuous or interval mode. This finding is consistent and clinically important, where usually symptoms that causes progressive losses related to mobility and functionality, and these predict decreases in quality of life and increase in mortality [27]. Studies have observed that usual exercise (i.e. walking endurance training) is recommended and considered a functional activity for people with PD [71]. In addition, individuals with PD invariably experience declines in balance and gait, and exercise can minimize these symptoms, improving physical functioning [72]. Therefore, delays in disease progression, which were higher in high-intensity training and correlate with functionality and independence, reinforce its effectiveness.

The meta-analysis for the cardiorespiratory fitness results showed a significant increase in the maximum oxygen consumption in individuals who trained at high intensity, both in the interval and continuous modality. Studies which related aerobic exercise with VO$_2$peak values corroborate our results, demonstrating that cardiopulmonary function and physical fitness improved rapidly after training in individuals with PD [70,73–75].

Our meta-analysis also showed a significant improvement in quality of life for those who engaged in high-intensity exercise when compared to those who did not perform any exercise, supporting the results of previously published meta-analysis [14,70,76,77]. Common comorbidities such as depression, functional and cognitive decline in PD, which gradually reduce individuals' quality of life, require training methods to support clinical treatment and reduce negative emotions caused by the disease [70].

Improvements related to the non-motor symptoms of PD were also evidenced after high-intensity training [21,38,43,44,78]. Although considered a movement disorder, non-motor manifestations such as autonomic dysfunction are also responsible for increasing the general burden of parkinsonian morbidity and mortality, further aggravating the clinical picture and limiting exercise performance, making the challenge of treating the disease even greater [2,79,80].

Endothelial reactivity, a non-motor symptom resulting from autonomic dysfunction and an important marker for assessing cardiovascular risk, improved significantly after high-intensity exercise performed in an interval manner (HIIT). This result probably suggests that alternating intensity during HIIT beneficially affected the shear stress in the arterial wall and produced better molecular responses, such as increased endothelial reactivity, since there were no improvements after exercise performed in this same study [25].

High-intensity exercise also demonstrated a greater impact on BDNF regulation than moderate-intensity exercise, increasing circulating BDNF levels after training [21,39,78]. These findings are important for prescribing training for people with neurodegenerative diseases, since BDNF is a factor that provides neuroprotection and cerebral neurodegeneration, serving

as a therapeutic agent, and improving the survival of dopaminergic neurons and directly influencing motor performance [81,82]. However, providing physical activity/ exercises in different intensities may be alternatives suggested in the literature to reduce all negative impacts in people with PD [83,84].

This systematic review has several limitations. A major limitation was the lack of details on adherence and adverse events in the included studies. Another limitation was differences and heterogeneity in the types of exercise programs, which included different modes of delivery and intensities. Different high-intensity types of exercises were reported across studies: continuous training, interval training, or a combination of both modalities. Methodological rigor was also a limiting factor, with some publications showing biases such as non-blinding of evaluators and incomplete results, which in turn confuse other researchers wishing to reproduce the study.

## Conclusion

In conclusion, this review has shown that high-intensity exercise performed in both continuous and interval modes may provide motor function benefits for individuals with Parkinson's disease. In this sense, it is observed that HIIT may be feasible, further studies are needed to establish the minimal consensus about patients with Parkinson disease can tolerate. The combination of high and moderate intensity may result in greater benefit and requires further investigation. Furthermore, it is important to mention that exercise should be an integral part of therapy for people with Parkinson's disease, always in conjunction with adjuvant treatment.

## Supporting information

**S1 Checklist. PRISMA 2020 checklist.**
(PDF)

**S1 Appendix. Appendix A.**
(DOCX)

**S2 Appendix. Appendix B.**
(DOCX)

## Author Contributions

**Conceptualization:** Ingrid Garcia De Sena, Maria Irany Knackfuss, Micássio Fernandes de Andrade, Paloma Katlheen Moura Melo, Ivana Alice Teixeira Fonseca.

**Data curation:** Ingrid Garcia De Sena, Isis Kelly dos Santos, Francisca Tayná da Silva Gomes, Paloma Katlheen Moura Melo.

**Formal analysis:** Ingrid Garcia De Sena, Isis Kelly dos Santos, Dayane Pessoa de Araújo, Francisca Tayná da Silva Gomes, Micássio Fernandes de Andrade, Paloma Katlheen Moura Melo.

**Investigation:** Adalberto Veronese da Costa, Dayane Pessoa de Araújo, José Rodolfo Lopes de Paiva Cavalcanti, Maria Irany Knackfuss, Ivana Alice Teixeira Fonseca.

**Methodology:** Ingrid Garcia De Sena, Adalberto Veronese da Costa, Isis Kelly dos Santos, Maria Irany Knackfuss, Micássio Fernandes de Andrade.

**Project administration:** Dayane Pessoa de Araújo, José Rodolfo Lopes de Paiva Cavalcanti.

**Resources:** Dayane Pessoa de Araújo, Maria Irany Knackfuss.

**Software:** Dayane Pessoa de Araújo, Francisca Tayná da Silva Gomes.

**Supervision:** Ivana Alice Teixeira Fonseca.

**Validation:** Isis Kelly dos Santos, José Rodolfo Lopes de Paiva Cavalcanti, Micássio Fernandes de Andrade, Ivana Alice Teixeira Fonseca.

**Visualization:** Adalberto Veronese da Costa, Isis Kelly dos Santos, Paloma Katlheen Moura Melo.

**Writing – original draft:** Ingrid Garcia De Sena, Adalberto Veronese da Costa, Isis Kelly dos Santos, Dayane Pessoa de Araújo, Francisca Tayná da Silva Gomes, José Rodolfo Lopes de Paiva Cavalcanti, Micássio Fernandes de Andrade, Paloma Katlheen Moura Melo, Ivana Alice Teixeira Fonseca.

**Writing – review & editing:** Adalberto Veronese da Costa, José Rodolfo Lopes de Paiva Cavalcanti, Maria Irany Knackfuss, Micássio Fernandes de Andrade, Ivana Alice Teixeira Fonseca.

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
