## [Decision Letter · Decision Letter 0]

15 Jun 2023

PONE-D-23-07053Feasibility and effect of high-intensity training on the progression of motor symptoms in adults individuals with Parkinson's disease: A systematic review and meta-analysisPLOS ONE

Dear Dr. Fonseca,

Thank you for submitting your manuscript to PLOS ONE. After careful consideration, we feel that it has merit but does not fully meet PLOS ONE’s publication criteria as it currently stands. Therefore, we invite you to submit a revised version of the manuscript that addresses the points raised during the review process.

Additional reviews have been obtained from reviewers with substantial expertise in the area of evaluating exercise impact in Parkinson’s disease in both human and rodent Parkinson’s models.   They see the value of this review from the perspective of high-intensity exercise.  This is a consensus I share with the reviewers.  The review has a number of strengths that include a summary of the risk of bias in cited studies.  However, some issues must be addressed in a revised manuscript.  These include the following points

1) Addressing a substantial amount of grammatical mistakes found throughout the manuscript.  An example is provided here from page 10

“However, there is no consensus for prescribing the intensity of exercise in PD, variations of training protocols can have different effects, changing the physiological response of individuals to exercise [26,27], studies indicate that in individuals with progressive neurodegenerative disease, when interferences appear that can modify the course of the disease and benefit these individuals, such as the practice of physical exercises, they need to be well informed [22]” 

2) Reviewers also mention that the authors should carefully delineate differences in delaying disease progression vs improving motor function.  Refer to reviewer comments for details.

3) The authors also must give more attention and comparison to outcomes obtained from moderate exercise practice in patients.  At present, most of the comparisons made to the high-intensity studies are “control groups”.      There are numerous studies of PD patients engaging in moderate intensity exercise with positive motor outcomes and some of these show positive motor outcomes from moderate exercise.  Here are but a few of such studies.

Shulman LM, et al.  2013. Randomized clinical trial of 3 types of physical exercise for patients with Parkinson disease. JAMA Neurol 1169 70, 183-190.

Ridgel AL, et al. 2009. Forced, not voluntary, exercise improves motor function in Parkinson’s disease patients. Neurorehabiol Neural Repair 23, 600-608.

Kanegusuku H, et al. 2021. Influence of motor impairment on exercise capacity and quality of life in patients with Parkinson disease. J Exerc Rehabil 17, 241-246.

Salvatore MF, et al., 2022. Establishing equivalent aerobic exercise parameters between early-stage Parkinson's disease and Pink1 knockout rats. J Parkinsons Dis 12: 1897-1915.

4) The authors should have a brief discussion on whether high-intensity exercise may pose a significant barrier for exercise participation, particularly with attention the disease stage, thus leaving moderate intensity as a viable option.

A revision addressing all of these concerns is needed to advance the manuscript further.

We look forward to receiving your revised manuscript.

Kind regards,

Michael Francis Salvatore

Academic Editor

PLOS ONE

Journal Requirements:

NO

4. Please ensure that you include a title page within your main document. You should list all authors and all affiliations as per our author instructions and clearly indicate the corresponding author.

5. Please include your tables as part of your main manuscript and remove the individual files. Please note that supplementary tables (should remain/ be uploaded) as separate "supporting information" files

Additional Editor Comments:

Please refer to above comments from Editor

Reviewers' comments:

Reviewer's Responses to Questions

**Comments to the Author**

1. Is the manuscript technically sound, and do the data support the conclusions?

Reviewer #1: Partly

Reviewer #2: Yes

2. Has the statistical analysis been performed appropriately and rigorously? 

Reviewer #1: N/A

Reviewer #2: Yes

3. Have the authors made all data underlying the findings in their manuscript fully available?

Reviewer #1: Yes

Reviewer #2: Yes

4. Is the manuscript presented in an intelligible fashion and written in standard English?

Reviewer #1: No

Reviewer #2: Yes

5. Review Comments to the Author

Reviewer #1: This is a very important topic and a comprehensive review of all high-intensity articles would be an important contribution to the literature. I did not do more than skim this article however because it is not written in clearly understandable English. I would be happy to review a revision after the authors have worked with someone to fully translate into English. There were some terms like "lower members" where I can make guesses about what the authors mean - but really have no idea and thus think it's likely best to revise for readability first prior to reviewers devoting more time on the details. I also noted when downloading the PDF that this manuscript has been under review before and prior reviewers were also concerned about English language readability--among other concerns with which I agreed (definition of high-intensity exercise).

Should the authors choose to revise and resubmit I would recommend including in the discussion some practical next step suggestions for real-world implementation of these findings. They acknowledge in the introduction that it's a confusing space because methodology so different -- but what common threads can we draw to produce clear real-world exercise guidelines? To some degree the Parkinson's Foundation has already done this (see their publicly available table for PROVIDERS re exercise prescription on their website; and there is an accompanying journal article either just published or under review).

Reviewer #2: June 11, 2023

This resubmitted manuscript by De sena et al is a review of the role of intense exercise in patients with PD. The authors responded to the initial critique, so it's unclear as to why an additional reviewer was needed for the manuscript. Regardless, this reviewer has a few additional comments.

1.) The grammar still needs to be improved.

2.) Abstract: in the very last line, the authors indicate that high intensity exercise can delay the progression of the disease. The authors present no compelling evidence for such an effect on the disease progression itself. This is an over statement of the findings. Intense exercise appears to improve motor function (perhaps even delaying motor dysfunction), but this is far different from delaying the progression of the disease. The holy grail in PD treatment is recovery in nigrostriatal dopamine levels/function. The authors present no data suggesting that there is any dopamine recovery with exercise and as far as this reviewer is aware, no such data are available in the human literature. In preclinical animal studies, our lab has clearly shown motor improvement that is totally independent of nigrostriatal dopamine recovery (Churchill et al, 2017, Sconce et al., 2015, Hood et al, 2016) using a recovery/restoration study design. Therefore, the authors need to eliminate the phrase, 'delay disease progression', not only in the abstract but also on pages 2 (4th paragraph), page 12 (first full paragraph), and page 13 (last paragraph).

3.) Page 9: lower 'members' functionality. The use of the word, members, is a bit odd. I assume the authors mean 'limb'? This should also be corrected at the bottom of page 11.

6. PLOS authors have the option to publish the peer review history of their article (what does this mean?). If published, this will include your full peer review and any attached files.

Reviewer #1: No

Reviewer #2: No

---

## [Author Response · Author response to Decision Letter 0]

30 Aug 2023

Response Letter,

Dear editor, we thank you for the excellent comments of the reviewers and for the opportunity we

have been given to revise the manuscript, making its quality better and within the standards

required by PLOS ONE. Below are the answers to each question:

Questions

Addressing a substantial amount of grammatical mistakes found throughout the manuscript. An

example is provided here from page 10

Answer: Thank you, we sent this manuscript to English review.

Reviewers also mention that the authors should carefully delineate differences in delaying disease

progression vs improving motor function. Refer to reviewer comments for details.

Answer: Thank you, we change it.

The authors also must give more attention and comparison to outcomes obtained from moderate

exercise practice in patients. At present, most of the comparisons made to the high-intensity

studies are “control groups”. There are numerous studies of PD patients engaging in moderate

intensity exercise with positive motor outcomes and some of these show positive motor outcomes

from moderate exercise. Here are but a few of such studies.

Shulman LM, et al. 2013. Randomized clinical trial of 3 types of physical exercise for patients

with Parkinson disease. JAMA Neurol 1169 70, 183-190.

Ridgel AL, et al. 2009. Forced, not voluntary, exercise improves motor function in Parkinson’s

disease patients. Neurorehabiol Neural Repair 23, 600-608.

Kanegusuku H, et al. 2021. Influence of motor impairment on exercise capacity and quality of

life in patients with Parkinson disease. J Exerc Rehabil 17, 241-246.

Salvatore MF, et al., 2022. Establishing equivalent aerobic exercise parameters between earlystage

Parkinson's disease and Pink1 knockout rats. J Parkinsons Dis 12: 1897-1915.

Answer: We appreciate this comment and we have added this paragraph and these recommended

references: However, moderate-intensity exercise may be a viable option for facilitating

adherence and improving exercise enjoyment for Parkinson’s disease[71–73]. In

literature low- or moderate-intensity exercise produced the greatest improvements in gait

speed and cardiovascular fitness in humans and model animals diagnosed with

Parkinson's disease [41,74,75].

The authors should have a brief discussion on whether high-intensity exercise may pose a

significant barrier for exercise participation, particularly with attention the disease stage, thus

leaving moderate intensity as a viable option.

Answer: Thank you for all suggestions. We added this information in discussion: In addition,

previous studies have found that high-intensity training (HIT) produces health improvements,

although the intensity required may be intimidating, which may not be sufficient to facilitate

intrinsic motivation for exercise adherence [71]. Exercise intensity is an important variable to

dose and individualize the exercise stimulus for specific participants. However, moderateintensity

exercise may be a viable option for facilitating adherence and improving exercise

enjoyment for Parkinson’s disease [71–73].

Reviewer #1

1. This is a very important topic and a comprehensive review of all high-intensity articles would

be an important contribution to the literature. I did not do more than skim this article however

because it is not written in clearly understandable English. I would be happy to review a revision

after the authors have worked with someone to fully translate into English. There were some terms

like "lower members" where I can make guesses about what the authors mean - but really have

no idea and thus think it's likely best to revise for readability first prior to reviewers devoting

more time on the details. I also noted when downloading the PDF that this manuscript has been

under review before and prior reviewers were also concerned about English language readability-

-among other concerns with which I agreed (definition of high-intensity exercise)

Answer: We regret this and ask for your understanding. This manuscript has been peer-reviewed.

In addition, we reviewed all studies and excluded six studies because they didn't report details

about HIIT.

2. Should the authors choose to revise and resubmit I would recommend including in the

discussion some practical next step suggestions for real-world implementation of these findings.

They acknowledge in the introduction that it's a confusing space because methodology so

different -- but what common threads can we draw to produce clear real-world exercise

guidelines? To some degree the Parkinson's Foundation has already done this (see their publicly

available table for PROVIDERS re exercise prescription on their website; and there is an

accompanying journal article either just published or under review).

Answer: Thank you. We added this information in discussion: In addition, previous studies have

found that high-intensity training (HIT) produces health improvements, although the intensity

required may be intimidating, which may not be sufficient to facilitate intrinsic motivation for

exercise adherence [71]. Exercise intensity is an important variable to dose and individualize the

exercise stimulus for specific participants. However, moderate-intensity exercise may be a viable

option for facilitating adherence and improving exercise enjoyment for Parkinson’s disease [71–

73].

Reviewer #2

1. This resubmitted manuscript by De sena et al is a review of the role of intense exercise in

patients with PD. The authors responded to the initial critique, so it's unclear as to why an

additional reviewer was needed for the manuscript. Regardless, this reviewer has a few additional

comments.

Answer: Thank you, we tried to improve this paper based on suggestions.

2. The grammar still needs to be improved.

Answer: Thank you, we sent this manuscript to English review.

3. Abstract: in the very last line, the authors indicate that high intensity exercise can delay the

progression of the disease. The authors present no compelling evidence for such an effect on the

disease progression itself. This is an over statement of the findings. Intense exercise appears to

improve motor function (perhaps even delaying motor dysfunction), but this is far different from

delaying the progression of the disease. The holy grail in PD treatment is recovery in nigrostriatal

dopamine levels/function. The authors present no data suggesting that there is any dopamine

recovery with exercise and as far as this reviewer is aware, no such data are available in the human

literature. In preclinical animal studies, our lab has clearly shown motor improvement that is

totally independent of nigrostriatal dopamine recovery (Churchill et al, 2017, Sconce et al., 2015,

Hood et al, 2016) using a recovery/restoration study design. Therefore, the authors need to

eliminate the phrase, 'delay disease progression', not only in the abstract but also on pages 2 (4th

paragraph), page 12 (first full paragraph), and page 13 (last paragraph).

Answer: Thank you, we agree, and it was removed in all paper.

4.Page 9: lower 'members' functionality. The use of the word, members, is a bit odd. I assume the

authors mean 'limb'? This should also be corrected at the bottom of page 11.

Answer: We agree, and it was change; the information was wrong.

---

## [Decision Letter · Decision Letter 1]

27 Sep 2023

PONE-D-23-07053R1Feasibility and effect of high-intensity training on the progression of motor symptoms in adults individuals with Parkinson's disease: A systematic review and meta-analysisPLOS ONE

Dear Dr. Fonseca,

Thank you for submitting your manuscript to PLOS ONE. After careful consideration, we feel that it has merit but does not fully meet PLOS ONE’s publication criteria as it currently stands. Therefore, we invite you to submit a revised version of the manuscript that addresses the points raised during the review process.

We look forward to receiving your revised manuscript.

Kind regards,

Michael Francis Salvatore

Academic Editor

PLOS ONE

Journal Requirements:

**Additional Editor Comments:**

The reviewers noted improvements from the original submission and responsivity to the previous reviews. However, I concur with the reviewer opinion that there are still a sufficient number of grammatical mistakes. These detract from an important topic in exercise research (the role of intensity in receiving benefits). I also concur with the reviewer that the conclusion still is too biased against the value of moderate intensity exercise. Authors are also encouraged to double check that all references listed in the bibliography are cited in the main text.

Reviewers' comments:

Reviewer's Responses to Questions

**Comments to the Author**

1. If the authors have adequately addressed your comments raised in a previous round of review and you feel that this manuscript is now acceptable for publication, you may indicate that here to bypass the “Comments to the Author” section, enter your conflict of interest statement in the “Confidential to Editor” section, and submit your "Accept" recommendation.

Reviewer #1: (No Response)

Reviewer #2: All comments have been addressed

2. Is the manuscript technically sound, and do the data support the conclusions?

Reviewer #1: Partly

Reviewer #2: (No Response)

3. Has the statistical analysis been performed appropriately and rigorously? 

Reviewer #1: I Don't Know

Reviewer #2: (No Response)

4. Have the authors made all data underlying the findings in their manuscript fully available?

Reviewer #1: Yes

Reviewer #2: (No Response)

5. Is the manuscript presented in an intelligible fashion and written in standard English?

Reviewer #1: No

Reviewer #2: (No Response)

6. Review Comments to the Author

Reviewer #1: The manuscript is improved from prior however still has some English language mistakes, incomplete sentences and overall would benefit tremendously from tighter writing. I get the gist of what they are trying to accomplish from the introduction.

The clarification of high vs moderate exercise and HIIT Is helpful. The tables are great and very useful. Likely the tables are the biggest contribution to the literature-it is helpful to have the studies summarized in such a way. But the conclusions drawn in the discussion are a bit far reaching/too broad. I defer to the editors of PLOSOne if this paper is up to standard for publishing in your journal. The content is important. But in my opinion the writing is not tight enough for publication in which you want to make it as easy as possible for your reader to seamlessly grasp the point.

Just a few specific examples as I was reading the paper:

This is not a full sentence: Based on symptoms, pharmacotherapy is the primary intervention for treating the

78 disease, but its adverse effects and reduced efficacy over time suggest implementing therapies

79 which modify the course of the disease and protect neurons against damage resulting from

80 neurodegeneration [8–10].

This sentence doesn’t make sense to me: However, the effectiveness of these exercises when

84 faced with delays in motor symptoms progressing to more advanced stages of the disease has

85 come to be questioned [9,15,16].

This is not a complete sentence: Studies suggest that if interventions (such as the practice of

106 physical exercise) which may modify the course of the disease and benefit individuals with

107 progressive neurodegenerative disease, they need to be well informed [28].

Our meta-analysis also showed a significant improvement in the quality of life of

434 individuals submitted to high intensity exercise when compared to those who did not perform

435 any exercise, � submitted is the wrong word here.

IN discussion would use word FEASIBLE rather than viable. Last sentence of first paragraph of discussion-I think they mean “adjuvant treatment to PHARMACOLOGICAL (not non-pharam) treatment”

Last sentence of conclusion in first person is completely unnecessary.

Reviewer #2: (No Response)

7. PLOS authors have the option to publish the peer review history of their article (what does this mean?). If published, this will include your full peer review and any attached files.

Reviewer #1: No

Reviewer #2: No

---

## [Author Response · Author response to Decision Letter 1]

6 Oct 2023

Thank you very much for reviewing our manuscript, and for providing helpful suggestions that strengthen our work. Below we address all the comments and questions and provide a summary after changes to the manuscript. In addition, please note that we reviewed all grammar issue and made a few changes. All suggestions were in red lines.

---

## [Editor Report · Decision Letter 2]

11 Oct 2023

Feasibility and effect of high-intensity training on the progression of motor symptoms in adults individuals with Parkinson's disease: A systematic review and meta-analysis

PONE-D-23-07053R2

Dear Dr. Fonseca,

We’re pleased to inform you that your manuscript has been judged scientifically suitable for publication and will be formally accepted for publication once it meets all outstanding technical requirements.

Kind regards,

Michael Francis Salvatore

Academic Editor

PLOS ONE
---

## [Editor Report · Acceptance letter]

17 Oct 2023

PONE-D-23-07053R2 

Feasibility and effect of high-intensity training on the progression of motor symptoms in adult individuals with Parkinson’s disease: A systematic review and meta-analysis 

Dear Dr. Fonseca:

I'm pleased to inform you that your manuscript has been deemed suitable for publication in PLOS ONE. Congratulations! Your manuscript is now with our production department. 

Kind regards, 

on behalf of

Dr. Michael Francis Salvatore 

Academic Editor

PLOS ONE